# Effect of Second-Hand Smoke Exposure on Establishing Urinary Cotinine-Based Optimal Cut-Off Values for Smoking Status Classification in Korean Adults

**DOI:** 10.3390/ijerph19137971

**Published:** 2022-06-29

**Authors:** Hyun-Seung Lee, Ji-Hyun Cho, Young-Jin Lee, Do-Sim Park

**Affiliations:** Department of Laboratory Medicine, School of Medicine, Wonkwang University, Iksan 54538, Korea; jihchojih@wku.ac.kr (J.-H.C.); jin20lab@wku.ac.kr (Y.-J.L.)

**Keywords:** cotinine, second-hand smoke, cut-off, the surveillance and monitoring

## Abstract

Regulations for banning smoking in indoor public places and workplaces have increased worldwide in recent years. A consecutive Korean National Health and Nutrition Examination Survey (KNHANES) between 2008 and 2018 showed a trend toward significant decreases in self-reported tobacco smoke exposure and measured urinary cotinine concentrations. We established and compared each optimal cut-off value for assessing the effect of second-hand smoke (SHS) exposure on establishing urinary cotinine-based cut-off values for smoking status classification in a population setting controlled for racial and cultural diversity, using four KNHANES datasets consisting of the 2008, 2011, 2014, and 2018 surveys. A total of 18,229 Korean participants aged >19 years with measured urinary cotinine concentrations were enrolled. Self-reports of current smoking status showed that the prevalence of current smokers decreased from 22.9% to 18.2% between 2008 and 2018. During this period, the median value of urinary cotinine in nonsmokers decreased from 5.86 µg/L to 0.48 µg/L, whereas the median value showed no remarkable decrease in current smokers. The AUC-based optimal cut-off values of urinary cotinine concentration for distinguishing current smokers from nonsmokers decreased from 86.5 µg/L to 11.5 µg/L. Our study showed that decreased SHS exposure would result in decreased optimal cut-off values for distinguishing current smokers from nonsmokers. In addition, the study suggests that the range of urinary cotinine concentration to define SHS exposure for the trend monitoring of populationof SHS exposure is appropriate between 0.30 µg/L and 100 µg/L. In addition, our study showed the importance of determination of cotinine concentration, which would have allowed us to avoid mistakes in qualification to the study group in an increased use of e-cigarette setting.

## 1. Introduction

Tobacco smoking is a common public health issue and a cause of preventable morbidity and mortality worldwide [1]. Smoking or exposure to second-hand smoke (SHS) is associated with the risk of asthma, chronic obstructive lung disease, respiratory tract infection, and various cancers [2,3,4,5]. Regulations and social consensus for banning smoking in indoor public places and workplaces are becoming ubiquitous in recent years [6,7,8,9]. The consecutive Korean National Health and Nutrition Examination Survey (KNHANES) between 2008 and 2018 showed a trend toward significant decreases in self-reported tobacco smoke exposure and measured urinary cotinine concentrations [10,11,12]. Cotinine is a major metabolite of nicotine, and it is used as a biomarker for tobacco smoke exposure [12]. Urinary cotinine is a noninvasive biomarker, and it has comparable diagnostic performance for tobacco smoke exposure to serum cotinine [13,14]. Therefore, urinary cotinine has been used in KNHANES for many years to validate reported smoking status and to monitor population exposure to tobacco over time [15].

The cut-off values of urinary cotinine concentration, established for smoking status classification, might be affected by various factors such as sex, age, pregnancy, nicotine-containing food intake, ethnic variation of cotinine half-life and nicotine metabolism, and SHS exposure [16,17,18,19]. Therefore, cut-off values of urinary cotinine concentration established by previous studies varied from 31.5 µg/L to 550 µg/L [18,20,21,22,23,24,25]. Although recent studies have reported that the cut-off values of cotinine have decreased over the past 20 years [25], the generalization of these trends is still limited without controlling the heterogeneity of racial and cultural factors related to nicotine absorption and metabolism.

In the current study, we established and compared each area under the curve (AUC)-based optimal cut-off value of urinary cotinine concentration using four KNHANES datasets from 2008, 2011, 2014, and 2018. We aimed to assess the effect of a declining trend in SHS exposure on establishing urinary cotinine-based cut-off values for smoking status classification during the recent decade in a controlled population setting with racial and cultural diversity.

## 2. Materials and Methods

### 2.1. Study Participants

A schematic flow chart of the study design and exclusion criteria are shown in Figure 1. Of the total of 33,804 participants who participated in the 2008, 2011, 2014, and 2018 KNHANES surveys, which were conducted by the Korea Centers for Disease Control and Prevention (Appendix A), 7595 participants aged <19 years were excluded. Of 26,209 adult participants, we excluded 7980 for whom urinary cotinine concentrations were not measured. In total, 18,229 Korean participants with measured urinary cotinine concentrations, aged ≥19 years, were enrolled in the current study. This study was approved by the Institutional Review Board of Wonkwang University Hospital (IRB file no. 2022-03-027-001). A waiver of consent was obtained given the nature of the project, which aimed to establish a cut-off value using a public dataset.

### 2.2. Self-Report for Smoking Status

In the current study, no response to smoking status was defined as a participant who did not participate in the self-report for smoking status or did not answer the self-report among all enrolled participants. Current smokers were defined as participants who reported a history of smoking 100 cigarettes in their lifetime and reported currently smoking cigarettes. A daily smoker was defined as a participant who reported, “Yes, I smoke at least one cigarette a day”. among current smokers. Non-daily smokers were defined as participants who reported “Yes, I smoke, but not every day”. among current smokers. Nonsmokers were defined as participants who did not meet the current smoking definition among participants for self-reported smoking status. The questionnaires for smoking status and data processing in the current study are described in Appendix A.

### 2.3. Self-Report for SHS Exposure

In the current study, SHS exposure was defined as a participant who reported a history of SHS exposure at home or in the workplace among nonsmokers. An unclear response to SHS exposure was defined as a participant who is not included in SHS exposure and who reported an unclear history of SHS exposure, such as no response, unknown response, and non-defined response, at home or in the workplace among nonsmokers. No SHS exposure was defined as a participant who definitively reported no history of SHS exposure at home and in the workplace among nonsmokers. The questionnaires for SHS exposure and data processing in the current study are described in Appendix A.

### 2.4. Self-Report for the Use of e-Cigarettes

In the current study, a current e-cigarette user was defined as a participant who reported a history of e-cigarette smoking within 30 days. A non-e-cigarette user was defined as a participant who definitively reported no history of e-cigarette smoking within 30 days. The questionnaires for the use of e-cigarettes and data processing in the current study are described in Appendix A.

### 2.5. Measurement of Urinary Cotinine Concentration

Measurement methods of urinary cotinine concentration and the diagnostic performance of the methods, including the limit of detection (LoD) and the limit of quantitation (LoQ), are described in Appendix A. Gas chromatography–tandem mass spectrometry (GC-MS/MS) with a Perkin Elmer Clarus 600T instrument (Perkin Elmer, Turku, Finland) was used in 2008, 2011, and 2014 to measure urinary cotinine concentrations. LoD of the GC-MS/MS method varied from 0.25 µg/L to 0.27 µg/L, and LoQ varied from 0.75 µg/L to 0.82 µg/L, between 2008 and 2014. In 2018, a high-performance liquid chromatography–tandem mass spectrometry (HPLC-MS/MS) system comprising an 1100 HPLC system (Agilent, Santa Clara, CA, USA) to API 4000 (AB Sciex, Redwood City, CA, USA) was used to measure urinary cotinine concentrations. LoD and LoQ of HPLC-MS/MS method were 0.27 µg/L and 0.31 µg/L, respectively.

### 2.6. Statistical Analysis

To reduce the impact of methodological differences on LoD in the current study, values equal to or lower than 0.30 µg/L were converted to 0.30 µg/L. Histograms of urinary cotinine concentration are expressed as log10 urinary cotinine (µg/L) and percentage frequency. Data with normal distribution are expressed as mean ± SD, while skewed data are expressed as median (interquartile range, IQR). However, some urinary cotinine concentration statistics associated with smoking status are intentionally expressed as mean ± SD because tailed portions in the distribution are important to establish cut-off values of urinary cotinine concentration for smoking status classification. The AUC-receiver operating characteristic (ROC) curve analysis with the maximum value of Youden’s index was used to establish optimal cut-off values of urinary cotinine concentration for smoking status classification. Fisher’s exact test or the chi-square test was used to analyze categorical data. The Mann–Whitney U test or Kruskal–Wallis test was used to analyze continuous data. The Statistical Package for the Social Sciences version 25.0 (IBM Corporation, Armonk, NY, USA) and the GraphPad Prism (version 9.1.2.; GraphPad Software, La Jolla, CA, USA) were used for statistical analyses and graphs. A *p* value of less than 0.05 was considered significant.

## 3. Results

### 3.1. The Characteristics of Study Participants

The characteristics of the study participants are described in Table 1. Total enrolled participant numbers in 2008, 2011, 2014, and 2018 were 5658, 1857, 4939, and 5775. The percentage of females was 55.4% (3135/5658), 48.0% (892/1857), 56.8% (2804/4939), and 55.0% (3179/5775) in each year, respectively. The mean ages of enrolled participants were 49.1 ± 16.3 years, 45.6 ± 14.7 years, 51.6 ± 16.4 years, and 49.1 ± 16.8 years, respectively. The mean values of urinary cotinine concentration were 314.6 ± 691.8 µg/L, 363.5 ± 761.3 µg/L, 286.4 ± 655.6 µg/L, and 277.7 ± 662.8 µg/L, respectively. Participants with equal to or less than 0.30 µg/L of urinary cotinine were 20.0% (1129/5658), 6.1% (114/1857), 4.1% (202/4939), and 21.1% (1216/5775), respectively.

### 3.2. The Result of the Self-Reported Smoking Status

The results of the self-reported smoking status in the current study are described in Table 1 and Table 2. The participants who responded to the self-report of smoking status in 2008, 2011, 2014, and 2018 were 99.4% (5627/5658), 98.7% (1832/1857), 93.9% (4639/4939), and 99.4% (5739/5775), respectively. Among them, current smokers were 22.9% (1287/5627), 24.9% (457/1832), 19.1% (885/4639), and 18.2% (1047/5739), respectively. Among the current smokers, daily smokers were 94.1% (430/457), 89.7% (794/885), and 93.6% (906/1047), in 2011, 2014, and 2018. Non-daily smokers were 5.9% (27/457), 10.3% (91/885), and 6.4% (141/1047). Only one was a non-daily smoker in 2008.

### 3.3. The Result of the Self-Reported SHS Exposure

The results of the self-reported SHS exposure in the current study are described in Table 3. The participants with SHS exposure in 2008, 2011, 2014, and 2018 were 37.1% (1610/4340), 34.9% (480/1375), 27.4% (1030/3754), and 6.8% (320/4692), respectively. No SHS exposure participants were 61.2% (2657/4340), 60.6% (833/1375), 72.5% (2722/3754), and 90.3% (4236/4692), respectively. The participants who submitted unclear self-reports of SHS exposure were 1.7% (73/4340), 4.5% (62/1375), 0.1% (2/3754), and 2.9% (136/4692), respectively. Of participants with equal to or less than 0.30 µg/L of urinary cotinine, no SHS exposure participants were 69.6% (780/1121), 71.2% (79/111), 85.9% (159/185), and 96.4% (1148/1191), respectively.

### 3.4. The Result of the Self-Reported Use of e-Cigarettes

The results of the self-reported use of e-cigarettes in the current study are described in Table 1 and Table 4. The participants with current use of e-cigarettes in 2014 and 2018 were 1.3% (65/4939) and 2.8% (163/5775), respectively. Among the current e-cigarette users, 22% (14/65, in 2014) and 12% (19/163, in 2018) had self-reported themselves as nonsmokers, whereas 78% (51/65) and 88% (144/163) had reported themselves as current smokers. The median values of urinary cotinine concentrations of the current e-cigarette users (1124.2 µg/L [711.6–1832.0 µg/L] in 2014; 1408.0 µg/L [740.0–2122.0 µg/L] in 2018) were comparable with those of daily smokers (1291.4 µg/L [779.7–1870.8 µg/L] in 2014; 1422.0 µg/L [820.0–1948.0 µg/L]).

### 3.5. The Distributions of Urinary Cotinine Concentration from All Participants

The distribution of urinary cotinine concentration in all participants is illustrated in Figure 2. Trimodal distribution of urinary cotinine concentration was observed based on a visual inspection of the histogram. The first subgroup was located at 0.30 µg/L. This subgroup was prominent in 2008 and 2018, whereas it decreased in 2011 and 2014. The second subgroup was located at various positions from 0.30 µg/L to around 100 µg/L in each survey, and this subgroup shifted toward 0.30 µg/L during the last decade. In 2018, the second subgroup showed a merged pattern with the first subgroup. The third subgroup started at approximately 100 µg/L and was most frequently observed at approximately 1500 µg/L. The third subgroup did not show markedly different locations or patterns among the four surveys.

### 3.6. The Distributions of Urinary Cotinine Concentration in Nonsmokers

The distribution of urinary cotinine concentration in nonsmokers is illustrated in Figure 3. A bimodal distribution of urinary cotinine concentration was observed based on the visual inspection of the histogram. One subgroup had a concentration of 0.30 µg/L. This subgroup was the same as the first subgroup in the distribution of urinary cotinine concentration in all participants. The proportions of no-SHS-exposure participants in this subgroup significantly increased from 69.6% (780/1121) to 96.4% (1148/1191) between 2008 and 2018. The other subgroup was located at various positions from 0.30 µg/L to around 100 µg/L in each survey. This subgroup was the same as the second subgroup in the distribution of urinary cotinine concentration in all participants. The proportions of no-SHS-exposure participants in this subgroup significantly increased from 58.3% (1877/3219) to 88.2% (3088/3501) between 2008 and 2018. However, the proportions of the first subgroup among no-SHS-exposure participants were 29.4% (780/2657) and 27.1% (1148/4236) in 2008 and 2018, while those were 9.5% (79/833) and 5.8% (159/2722) in 2011 and 2014. The proportions of the second subgroup among no SHS exposure participants were 70.6% (1877/2657) and 72.9% (3088/4236) in 2008 and 2018, while those were 90.5% (754/833) and 94.2% (2563/2722) in 2011 and 2014. The third subgroup of distribution for urinary cotinine values from all participants was almost undetectable in nonsmokers.

### 3.7. The Established Optimal Cut-Off Values of Urinary Cotinine Concentration for Smoking Status Classification

The established optimal cut-off values of urinary cotinine concentration for smoking status classification are described in Table 5 and illustrated in in Figure 2. According to smoking status, two cut-off values were established to distinguish current smokers from nonsmokers and to distinguish daily smokers from non-daily smokers. The optimal cut-off values for distinguishing current smokers from nonsmokers in 2008, 2011, 2014, and 2018 were 86.48 µg/L (95% confidence interval, 71.30–104.60 µg/L), 43.85 µg/L (38.51–67.70 µg/L), 15.93 µg/L (10.45–42.41 µg/L), and 11.50 µg/L (5.89–19.90 µg/L). The optimal cut-off values for distinguishing daily smokers from non-daily smokers in 2011, 2014, and 2018 were 107.91 µg/L (49.56–130.64 µg/L), 110.51 µg/L (79.71–135.29 µg/L), and 77.90 µg/L (11.53–112.00 µg/L). Only the optimal cut-off value of urinary cotinine concentration for distinguishing current smokers from nonsmokers was established in the 2008 KNHANES dataset. The diagnostic performance for each optimal cut-off varied from 93.00% to 98.85% for sensitivity and 94.06% to 96.29% for specificity.

### 3.8. Comparison of Diagnostic Performance According to Various Urinary Cotinine Cut-Off Values for Smoking Status Classification

A comparison of diagnostic performance according to various cut-off values of urinary cotinine concentration for smoking status classification is presented in Table 6 and illustrated in Figure 4. When using 100 µg/L, 50 µg/L, 25µg/L, and 12 µg/L as cut-off values for distinguishing current smokers from nonsmokers, sensitivity and negative predictive value (NPV) were relatively similar among the four datasets. As the cut-off value, 12 µg/L showed the best performance for sensitivity (96.06% to 98.91%) and NPV (98.54% to 99.71%), whereas 100 µg/L showed the worst performance for sensitivity (93.22% to 96.85%) and NPV (97.71% to 99.28%). In contrast, the specificity and positive predictive value (PPV) were significantly different according to the survey. As the cut-off value, 12 µg/L showed the worst performance for specificity and PPV, whereas 100 µg/L showed the best performance. Specificities of 12 µg/L in 2008, 2011, 2014, and 2018 were 62.95% (61.49%–64.39%), 88.36% (86.55%–90.01%), 94.73% (93.96%–95.42%), and 95.82% (95.21%–96.38%). PPV were 44.19% (43.22%–45.16%), 73.29% (70.32%–76.06%), 81.53% (79.40%–83.49%), and 84.07% (82.14%–85.82%). However, the specificity and PPV of 100 µg/L ranged from 94.49% to 96.40%, and from 83.88% to 85.71%, respectively.

According to a visual inspection of the scatter plot of urinary cotinine concentration, the distribution patterns in current smokers were relatively similar among the four datasets. Most current smokers showed values greater than 100 µg/L of urinary cotinine, and only 2.0% (20/1047, in 2018) to 2.9% (13/457, in 2011) of current smokers showed values ranging from 12 µg/L to 100 µg/L in all datasets (Table 7). However, the distribution of nonsmokers showed a markedly decreased pattern during the recent decade. The percent of nonsmokers with values in the 12 µg/L to 100 µg/L range decreased from 31.5% (1369/4340) of nonsmokers in 2008 to 0.7% (27/4692) of nonsmokers in 2018.

### 3.9. Comparison between Self-Reported SHS Exposure and Urinary Cotinine Concentrations in Nonsmokers

A comparison between self-reported SHS exposure and urinary cotinine concentrations in nonsmokers is presented in Table 3. A decreasing trend in urinary cotinine values in nonsmokers was clearly observed between the surveys. The median values of urinary cotinine concentration in nonsmokers in 2008, 2011, 2014, and 2018 are 5.86 µg/L (0.30–20.03 µg/L), 2.83 µg/L (1.25–6.17 µg/L), 1.14 µg/L (0.67–2.19 µg/L), and 0.48 µg/L (0.30–0.84 µg/L), respectively.

## 4. Discussion

To define SHS exposure, the Centers for Disease Control and Prevention in the United States have used the range of serum cotinine between 0.05 µg/L and 10 µg/L [26]. A serum cotinine value of 0.05 µg/L is equal to the previous LoD of the LC-MS/MS method, and 10 µg/L is widely accepted as the cut-off value for defining smokers [26]. However, there is no consensus on the cut-off value of urinary cotinine concentration to define SHS exposure. A previous study using KNHANES datasets between 2007 and 2010 asserted that urinary cotinine concentration did not distinguish SHS exposure from nonsmokers because the half-life of cotinine is less than 24 h [15]. However, this conclusion might be misleading due to limitations in self-reporting for SHS exposure. Self-report of SHS exposure is subjective and less correlated with the quantitative values of urinary cotinine concentration. Different proportions of self-reported responses to SHS exposure and the proportion of unclear SHS exposure between the surveys may bias the correlation between self-reported SHS exposure and urinary cotinine values in nonsmokers. Our data clearly showed the limitations of self-reporting SHS exposure (Table 3). For example, the no-SHS-exposure participants among the participants with equal to or less than 0.30 µg/L in 2008 were 69.6% (780/1121). This is similar to the 71.2% (79/111) of the no SHS exposure participants among the participants with equal to or less than 0.30 µg/L in 2011, although the proportion of the participants with equal to or less than 0.30 µg/L differed significantly (25.8% [1121/4340] vs. 8.1% [111/1375], *p* < 0.0001). Furthermore, 28.5% (320/1121) of participants with equal to or less than 0.30 µg/L in 2008 reported SHS exposure, whereas most participants with more than 0.30 µg/L in 2018 reported no SHS exposure. Above all, the median value of urinary cotinine concentration from no-SHS-exposure participants in 2008 was significantly higher than that of SHS exposure participants, in 2011, 2014, and 2018 KNHANES. Therefore, the cut-off value of urinary cotinine concentration to define SHS exposure should be established based on the distribution of urinary cotinine concentrations rather than self-reports. Our study indicated that a 0.30 µg/L urinary cotinine value, which is close to the current LoD of GC-MS/MS or HPLC MS/MS, is the cut-off value for distinguishing SHS exposure from nonsmokers in the Korean adult population.

During the last decade, the use of novel forms of smoking, such as electronic cigarettes, heated tobacco products, little cigars, and cigarillos, has rapidly increased among the public. However, the urinary cotinine concentration distribution in current smokers does not appear to have changed markedly during this period in the current study (Table 7 and Figure 4). A study of urinary cotinine concentrations in e-cigarette users reported that e-cigarette users are exposed to the same nicotine levels as cigarette smokers [27], and the results of our study are in line with those of a previous study (Table 4, Appendix A). Therefore, the optimal cut-off values for distinguishing daily smokers from non-daily smokers did not change markedly at approximately 100 µg/L in each survey. The range of urinary cotinine concentration between 0.30 µg/L and 100 µg/L appears to be suitable for monitoring the trend of SHS exposure. These cut-off points, consisting of the LoD value for urinary cotinine concentration measurement and the cut-off value to define daily smokers, are similar to the characteristics of cut-point values of serum cotinine for SHS exposure, as defined by the Centers for Disease Control and Prevention in the United States.

Our study also showed that the optimal cut-off value for distinguishing current smokers from nonsmokers changed mainly because of the continuous quantitative decline of urinary cotinine concentration in participants with SHS exposure. Among the nonsmokers, the number of participants with the range of urinary cotinine concentration between 12 µg/L and 100 µg/L significantly decreased from 31.5% (1369/4340) in 2008 to 0.7% (27/4692) in 2018, whereas the current smokers with that range were not different, with 2.2% (26/1287) in 2008 and 2.0% (20/1047) in 2018 (Table 7). As a result, the optimal cut-off value for differentiating non-daily smokers and participants with SHS exposure decreased from 86.48 µg/L in 2008 to 11.50 µg/L in 2018. The optimal cut-off value of 11.50 µg/L for distinguishing current smokers from nonsmokers for the 2018 dataset showed equal or superior diagnostic performance compared to the optimal cut-off value of 86.48 µg/L for 2008 dataset (sensitivity 98.85% vs. 97.20%, specificity 95.80% vs. 94.06%; Table 5).

A transference of the cut-off value is generally unacceptable when there are significant geographic or demographic differences between the populations that are known to cause a difference in the cut-off value [28]. Furthermore, our study showed that the optimal cut-off value for distinguishing current smokers from nonsmokers could change according to the quantitative alteration of urinary cotinine concentration in nonsmokers, suggesting the intensity or frequency of SHS exposure, even in a controlled population with racial and cultural diversity. Therefore, it is ideal to investigate and analyze the pattern and characteristics of SHS exposure in different populations and then establish the population’s own cut-off value rather than a transference of the cut-off value. If transference of the cut-off value is the only option without a survey for SHS exposure, 100 µg/L might be recommended rather than other lower values of urinary cotinine concentration as the cut-off value for distinguishing current smokers from nonsmokers. Although improved sensitivity and NPV would be helpful for the individual management of smoking-related issues, cut-off values lower than 100 µg/L of urinary cotinine concentration may cause significantly lower specificities and PPVs in a high-SHS-exposure setting such as the 2008 KNHANES (Table 6).

Our study has some limitations. First, the smoking status verification was not performed by determination of cotinine concentration in serum, which would have allowed us to avoid mistakes in qualification to the study group. Next, several smoking statuses were ignored in the current study, such as former smokers, users of novel forms of smoking or nicotine alternatives, and participants undergoing nicotine replacement therapy. Of current e-cigarette users, 22% (14/65, in 2014) and 12% (19/163, in 2018) had self-reported as nonsmokers, and 0.5% (17/3754, in 2014) and 0.4% (19/4692, in 2018) of misclassified nonsmokers were identified, respectively. However, between 3.6% and 5.5% of suspected nonsmokers with 100 µg/L or higher value of urinary cotinine were not excluded in each survey (Table 7), because questionnaires for use of novel forms of smoking or nicotine alternatives were only included in two surveys, of the total surveys. These limitations might lead to a minor bias for statistics and established cut-off values, although they could not affect trends toward significant declines in measured urinary cotinine in nonsmokers and a decline in the optimal cut-off value for distinguishing current smokers from nonsmokers.

## 5. Conclusions

In conclusion, our study showed that decreased SHS exposure mainly resulted in decreased optimal cut-off values for distinguishing current smokers from nonsmokers in a controlled Korean population with racial and cultural diversity. Moreover, the current study suggests that the range of urinary cotinine concentration to define SHS exposure for population-monitoring the trend of SHS exposure is appropriate between 0.30 µg/L and 100 µg/L. In addition, our study showed the importance of determination of cotinine concentration, which would have allowed us to avoid mistakes in qualification to the study group in an increased use of e-cigarette setting.

## Figures and Tables

**Figure 1 ijerph-19-07971-f001:**
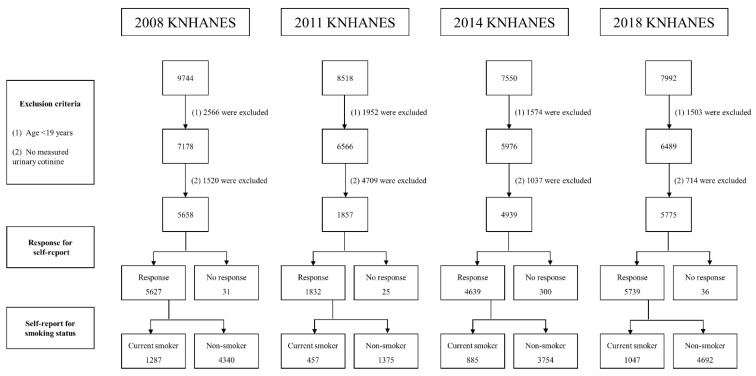
Schematic flow chart of study design and exclusion criteria in the current study.

**Figure 2 ijerph-19-07971-f002:**
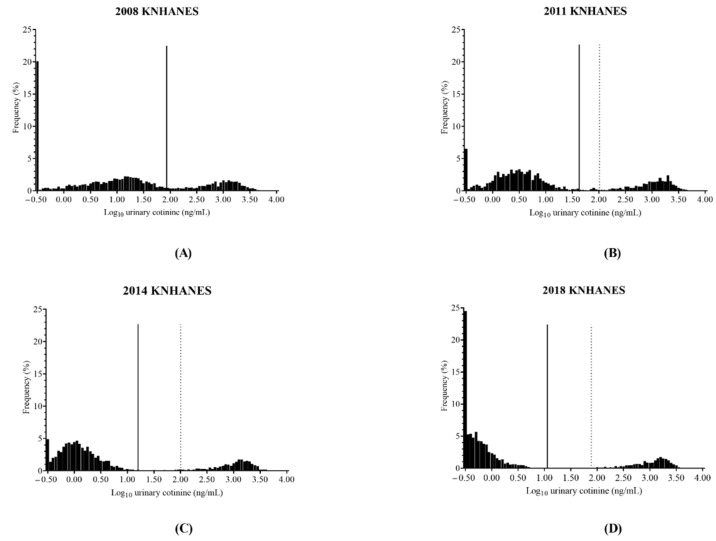
The distributions of urinary cotinine concentration from total participants and established optimal cut-off values for smoking status classification. 2008 KNHANES (**A**), 2011 KNANES (**B**), 2014 KNANES (**C**), and 2018 KNANES (**D**). The vertical solid lines represent the optimal cut-off values of urinary cotinine concentration to distinguish current smokers from nonsmokers in each dataset. The vertical dotted lines represent the optimal cut-off values of urinary cotinine concentration to distinguish daily and non-daily smokers in each dataset. Only the optimal cut-off value of urinary cotinine concentration for distinguishing current smokers from nonsmokers was established in the 2008 KNHANES dataset. KNHANES = Korean National Health and Nutrition Examination Survey.

**Figure 3 ijerph-19-07971-f003:**
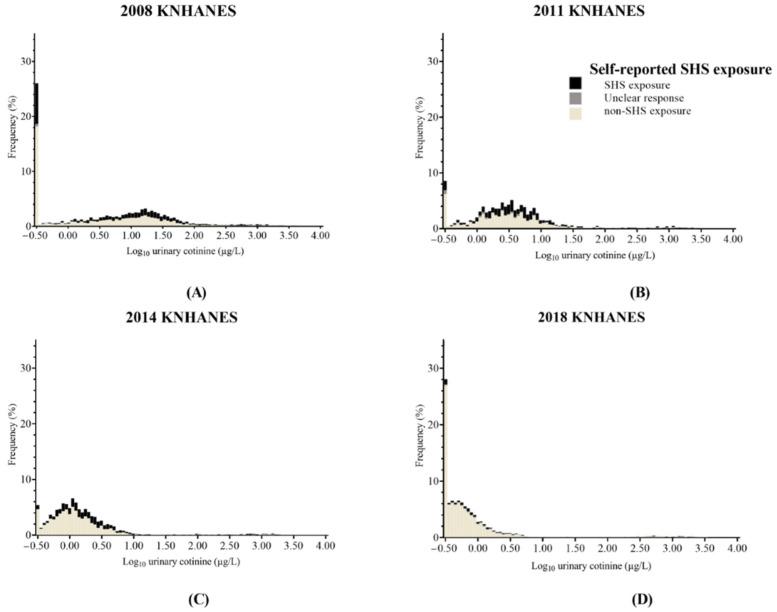
The distributions of urinary cotinine concentration from nonsmokers according to SHS exposure. 2008 KNHANES (**A**), 2011 KNANES (**B**), 2014 KNANES (**C**), and 2018 KNANES (**D**). The black, gray, and ivory boxes represent the percent frequency of SHS exposure, unclear response for SHS exposure, and no SHS exposure, respectively, among nonsmokers. KNHANES = Korean National Health and Nutrition Examination Survey. SHS = second-hand smoke.

**Figure 4 ijerph-19-07971-f004:**
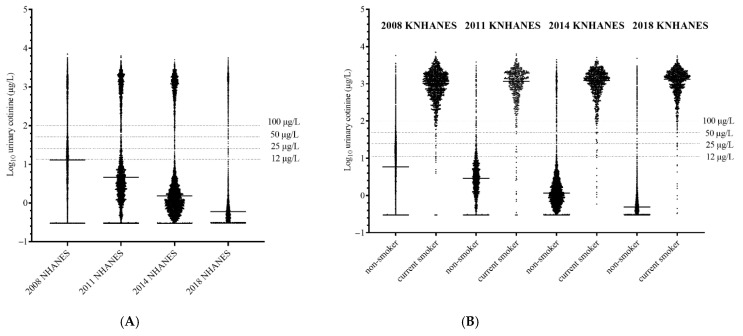
The comparison for distribution of urinary cotinine concentration in total enrolled participants (**A**) and according to smoking status classification (**B**) between 2008 and 2018 KNHANES datasets. The horizontal dotted lines represent the various cut-off values of urinary cotinine concentration for distinguishing current smokers from nonsmokers. The horizontal solid lines represent the median urinary cotinine values in each population. KNHANES = Korean National Health and Nutrition Examination Survey.

**Table 1 ijerph-19-07971-t001:** The characteristics of study participants.

Data Characteristics	KNHANES (2008–2018)			
	2008	2011	2014	2018
Total enrolled participants (n)	5658	1857	4939	5775
Age, year (mean ± SD)	49.1 ± 16.3	45.6 ± 14.7	51.6 ± 16.4	51.5 ± 16.8
Sex, female (n, %)	3135 (55.4%)	892 (48.0%)	2804 (56.8%)	3179 (55.0%)
Urinary cotinine (µg/L)	314.6 ± 691.8	363.5 ± 761.3	286.4 ± 655.6	277.7 ± 662.8
Response for self-report (n, %)	5627 (99.4%)	1832 (98.7%)	4639 (93.9%)	5739 (99.4%)
Nonsmoker, self-report (n, %)	4340 (77.1%)	1375 (75.1%)	3754 (80.9%)	4692 (81.8%)
Current smoker, self-report (n, %)	1287 (22.9%)	457 (24.9%)	885 (19.1%)	1047 (18.2%)
Use of e-cigarettes				
Current e-cigarette user (n, %)			65 (1.3%)	163 (2.8%)
Non-e-cigarette user (n, %)			4874 (98.7%)	5612 (97.2%)

KNHANES = Korean National Health and Nutrition Examination Survey. GC-MS/MS = gas chromatography–tandem mass spectrometry. HPLC-MS/MS = high-performance liquid chromatography–tandem mass spectrometry.

**Table 2 ijerph-19-07971-t002:** The comparison between self-report for daily smoking status and urinary cotinine in current smokers.

Year	Daily Smoking Status	≤100 µg/L of Urinary Cotinine		>100 µg/L of Urinary Cotinine		Total	
		n (%)	Concentration (µg/L, [Median, IQR])	n (%)	Concentration (µg/L)	n (%)	Concentration (µg/L)
2008	Daily smoker	43 (100.0%)	39.5 (9.7–76.3)	1243 (100.0%)	1067.1 (573.6–1712.9)	1286 (100.0%)	1032.1 (529.7–1687.6)
	Non-daily smoker			1	313.0	1	313.0
	total	43 (100.0%)	39.5 (9.7–76.3)	1244 (100.0%)	1067.1 (573.6–1712.9)	1287 (100.0%)	1032.1 (529.7–1687.6)
2011	Daily smoker	19 (61.3%)	2.9 (1.1–14.9)	411 (96.5%)	1304.3 (728.2–1975.1)	430 (94.1%)	1251.0 (678.9–1945.0)
	Non-daily smoker	12 (38.7%)	33.3 (7.5–75.6)	15 (3.5%)	423.2 (205.2–1371.7)	27 (5.9%)	107.9 (45.9–425.1)
	total	31 (100.0%)	5.7 (1.2–49.0)	426 (100.0%)	1275.3 (710.1–1962.6)	457 (100.0%)	1170.4 (559.1–1935.1)
2014	Daily smoker	6 (17.6%)	55.1 (0.9–84.0)	788 (92.6%)	1310.7 (808.5–1886.2)	794 (89.7%)	1295.7 (797.4–1876.6)
	Non-daily smoker	28 (83.4%)	30.0 (7.9–49.5)	63 (7.4%)	548.3 (254.5–1015.6)	91 (10.3%)	310.5 (64.5–758.6)
	total	34 (100.0%)	27.5 (4.9–67.3)	851 (100.0%)	1257.1 (730.6–1844.1)	885 (100.0%)	1231.0 (670.6–1823.6)
2018	Daily smoker	8 (22.8%)	42.2 (8.8–84.3)	898 (88.7%)	1438.0 (844.0–1980.0)	906 (93.6%)	1434.0 (828.3–1973.0)
	Non-daily smoker	27 (77.2%)	21.8 (2.1–69.1)	114 (11.3%)	577.5 (335.0–1228.0)	141 (6.4%)	472.0 (174.5–1069.0)
	total	35 (100.0%)	24.0 (2.1–74.0)	1012 (100.0%)	1374.0 (765.5–1908.0)	1047 (100.0%)	1324.0 (697.5–1890.0)

**Table 3 ijerph-19-07971-t003:** The comparison between self-report for SHS exposure and urinary cotinine value in nonsmokers.

Year	≤0.30 µg/L of Urinary Cotinine (%)	Self-Report for SHS Exposure	≤0.30 µg/L of Urinary Cotinine		>0.30 µg/L of Urinary Cotinine		Total	
			n (%)	Concentration (µg/L)	n (%)	Concentration (µg/L, [Median, IQR])	n (%)	Concentration (µg/L)
2008	25.8%	SHS exposure	320 (28.5%)	0.30	1290 (40.1%)	15.03 (4.97–35.02)	1610 (37.1%)	9.54 (1.34–26.97)
		Unclear response	21 (1.9%)	0.30	52 (1.6%)	8.40 (1.65–19.05)	73 (1.7%)	2.26 (0.30–11.46)
		No SHS exposure	780 (69.6%)	0.30	1877(58.3%)	10.40 (3.47–22.98)	2657 (61.2%)	4.26 (1.00–16.47)
		Total	1121 (100.0%)	0.30	3219 (100.0%)	11.95 (3.98–26.91)	4340 (100.0%)	5.86 (0.30–20.03)
2011	8.1%	SHS exposure	22 (19.8%)	0.30	458 (36.2%)	3.53 (1.91–7.39)	480 (34.9%)	3.33 (1.61–7.15)
		Unclear response	10 (9.0%)	0.30	52 (4.1%)	2.81 (1.19–4.36)	62 (4.5%)	1.78 (0.64–4.12)
		No SHS exposure	79 (71.2%)	0.30	754 (59.7%)	2.94 (1.51–6.29)	833 (60.6%)	2.54 (1.16–5.45)
		Total	111 (100.0%)	0.30	1264 (100.0%)	3.20 (1.59–6.64)	1375 (100.0%)	2.83 (1.25–6.17)
2014	4.9%	SHS exposure	26 (14.1%)	0.30	1004 (28.1%)	1.67 (0.93–3.29)	1030 (27.4%)	1.61 (0.88–3.21)
		Unclear response			2 (0.1%)	1.85 (0.66–3.03)	2 (0.1%)	1.85 (0.66–3.03)
		No SHS exposure	159 (85.9%)	0.30	2563 (71.8%)	1.09 (0.68–1.91)	2722 (72.5%)	1.03 (0.62–1.84)
		Total	185 (100.0%)	0.30	3569 (100.0%)	1.20 (0.73–2.28)	3754 (100.0%)	1.14 (0.67–2.19)
2018	25.4%	SHS exposure	36 (3.3%)	0.30	284 (8.1%)	0.82 (0.54–1.95)	320 (6.8%)	0.73 (0.44–1.43)
		Unclear response	7 (0.3%)	0.30	129 (3.7%)	1.23 (0.67–2.68)	136 (2.9%)	1.21 (0.62–2.54)
		No SHS exposure	1148 (96.4%)	0.30	3088 (88.2%)	0.59 (0.42–0.99)	4236 (90.3%)	0.46 (0.30–0.78)
		Total	1191 (100.0%)	0.30	3501 (100.0%)	0.62 (0.43–1.07)	4692 (100.0%)	0.48 (0.30–0.84)

SHS = second-hand smoke.

**Table 4 ijerph-19-07971-t004:** The comparison between self-report for use of e-cigarettes and urinary cotinine concentration in self-report participants.

Year	Self-Report for SHS Exposure	Current e-Cigarette Users		Non-e-Cigarette Users		Total	
		n (%)	Concentration (µg/L, [Median, IQR])	n (%)	Concentration (µg/L)	n (%)	Concentration (µg/L)
2014	Daily smoker	47 (72.3%)	1473.9 (973.3–2015.6)	747 (15.3%)	1291.4 (779.7–1870.8)	794 (16.1%)	1295.7 (797.4–1876.6)
	Non-daily smoker	4 (6.2%)	844.7 (45.6–1394.5)	87 (1.8%)	310.5 (64.5–735.6)	91 (2.9%)	310.5 (64.5–758.6)
	Nonsmoker with SHS exposure	6 (9.2%)	2.8 (2.4–676.5)	1024 (21.0%)	1.61 (0.88–3.21)	1030 (20.9%)	1.61 (0.88–3.21)
	Nonsmoker with unclear response			2 (0.1%)	1.85 (0.66–3.03)	2 (0.0%)	1.85 (0.66–3.03)
	Nonsmoker without SHS exposure	8 (12.3%)	760.4 (57.7–1175.0)	2714 (55.7%)	1.03 (0.62–1.82)	2722 (55.1%)	1.03 (0.62–1.84)
	total	65 (100.0%)	1124.2 (711.6–1832.0)	4874 (100.0%)	1.48 (0.75–6.14)	4939 (100.0%)	1.52 (0.76–7.22)
2018	Daily smoker	124 (76.1%)	1540.0 (870.0–2176.0)	782 (14.0%)	1422.0 (820.0–1948.0)	906 (16.1%)	1434.0 (828.3–1973.0)
	Non-daily smoker	20 (12.3%)	693.5 (424.5–1084.5)	121 (2.2%)	352.0 (96.9–906.5)	141 (2.5%)	472.0 (174.5–1069.0)
	Nonsmoker with SHS exposure	5 (3.1%)	1530.0 (1068.0–1884.0)	318 (5.7%)	0.73 (0.30–1.54)	320 (5.7%)	0.73 (0.44–1.43)
	Nonsmoker with unclear response	1 (0.6%)	2168	135 (2.4%)	1.21 (0.62–2.53)	136 (2.4%)	1.21 (0.62–2.54)
	Nonsmoker without SHS exposure	13 (8.0%)	712.0 (0.43–1528.0)	4223 (75.7%)	0.46 (1.08–0.78)	4236 (75.5%)	0.46 (0.30–0.78)
	total	163 (100.0%)	1408.0 (740.0–2122.0)	5576 (100.0%)	0.58 (0.33–17.75)	5612 (100.0%)	1.48 (0.75–6.14)

SHS = second-hand smoke.

**Table 5 ijerph-19-07971-t005:** The established optimal cut-off values of urinary cotinine concentration for smoking status classification.

Year	Classification	Optimal Cut-Off Value of Urinary Cotinine (µg/L)	Sensitivity (%)	Specificity (%)	Youden’s Index	AUC	*p* Value
2008	Current smoker	86.48	97.20	94.06	0.913	0.978	<0.0001
		(95% CI, 71.30–104.60)	(96.15–98.03)	(93.31–94.74)	(0.899–0.922)	(0.974–0.982)	
	Daily smoker						
2011	Current smoker	43.85	95.19	94.84	0.901	0.962	<0.0001
		(38.51–67.70)	(92.80–96.96)	(93.53–95.95)	(0.873–0.921)	(0.952–0.970)	
	Daily smoker	107.91	93.00	96.15	0.906	0.962	<0.0001
		(49.56–130.64)	(90.26–95.16)	(94.99–97.10)	(0.881–0.925)	(0.953–0.971)	
2014	Current smoker	15.93	98.53	95.05	0.936	0.983	<0.0001
		(10.45–42.41)	(97.50–99.22)	(94.30–95.72)	(0.923–0.944)	(0.978–0.986)	
	Daily smoker	110.51	95.99	96.16	0.933	0.980	<0.0001
		(79.71–135.29)	(94.51–97.16)	(95.50–96.76)	(0.923–0.941)	(0.975–0.983)	
2018	Current smoker	11.50	98.85	95.80	0.947	0.986	<0.0001
		(5.89–19.90)	(98.01–99.41)	(95.19–96.36)	(0.936–0.954)	(0.982–0.989)	
	Daily smoker	77.90	97.33	96.29	0.933	0.981	<0.0001
		(11.53–112.00)	(96.16–98.22)	(95.71–96.81)	(0.922–0.940)	(0.978–0.985)	

AUC = area under the curve. CI = confidence interval.

**Table 6 ijerph-19-07971-t006:** The comparison of diagnostic performance according to various cut-off values of urinary cotinine concentration for distinguishing current smokers from nonsmokers.

Year	Current Smoking Prevalence (%)	Cut-Off Value of Urinary Cotinine (µg/L)	Sensitivity (%)	Specificity (%)	PPV (%)	NPV (%)	AUC	*p* Value
2008	22.87%	12	98.91	62.95	44.19	99.49	0.809	<0.0001
			(98.18 to 99.40)	(61.49 to 64.39)	(43.22 to 45.16)	(99.14 to 99.70)	(0.799 to 0.819)	
		25	98.52	79.98	59.34	99.46	0.893	<0.0001
			(97.70 to 99.11)	(78.76 to 81.16)	(57.88 to 60.77)	(99.15 to 99.65)	(0.884 to 0.900)	
		50	97.98	90.39	75.15	99.34	0.942	<0.0001
			(97.05 to 98.68)	(89.48 to 91.25)	(73.40 to 76.82)	(99.04 to 99.55)	(0.935 to 0.948)	
		100	96.66	94.49	83.88	98.96	0.956	<0.0001
			(95.53 to 97.57)	(93.77 to 95.15)	(82.14 to 85.49)	(98.61 to 99.23)	(0.950 to 0.961)	
2011	24.95%	12	96.06	88.36	73.29	98.54	0.922	<0.0001
			(93.85 to 97.65)	(86.55 to 90.01)	(70.32 to 76.06)	(97.72 to 99.07)	(0.909 to 0.934)	
		25	95.19	93.24	82.39	98.31	0.942	<0.0001
			(92.80 to 96.96)	(91.78 to 94.51)	(79.34 to 85.07)	(97.48 to 98.87)	(0.930 to 0.952)	
		50	94.97	94.98	86.28	98.27	0.950	<0.0001
			(92.54 to 96.78)	(93.69 to 96.08)	(83.31 to 88.79)	(97.44 to 98.83)	(0.939 to 0.959)	
		100	93.22	96.07	88.75	97.71	0.946	<0.0001
			(90.51 to 95.35)	(94.91 to 97.04)	(85.85 to 91.12)	(96.81 to 98.36)	(0.935 to 0.956)	
2014	19.08%	12	98.76	94.73	81.53	99.69	0.967	<0.0001
			(97.79 to 99.38)	(93.96 to 95.42)	(79.40 to 83.49)	(99.45 to 99.83)	(0.962 to 0.972)	
		25	98.31	95.21	82.86	99.58	0.968	<0.0001
			(97.22 to 99.05)	(94.47 to 95.87)	(80.73 to 84.79)	(99.31 to 99.75)	0.962 to 0.972	
		50	97.18	95.55	83.74	99.31	0.964	<0.0001
			(95.86 to 98.16)	(94.84 to 96.19)	(81.61 to 85.66)	(98.99 to 99.53)	(0.958 to 0.969)	
		100	96.16	96.08	85.27	99.07	0.961	<0.0001
			(94.67 to 97.33)	(95.41 to 96.68)	(83.16 to 87.16)	(98.71 to 99.33)	(0.955 to 0.967)	
2018	18.24%	12	98.76	95.82	84.07	99.71	0.973	<0.0001
			(97.89 to 99.34)	(95.21 to 96.38)	(82.14 to 85.82)	(99.51 to 99.83)	(0.968 to 0.977)	
		25	98.28	96.01	84.62	99.60	0.971	<0.0001
			(97.30 to 98.98)	(95.42 to 96.56)	(82.70 to 86.37)	(99.37 to 99.75)	(0.967 to 0.976)	
		50	97.89	96.10	84.85	99.51	0.970	<0.0001
			(96.84 to 98.68)	(95.51 to 96.64)	(82.93 to 86.59)	(99.27 to 99.68)	(0.965 to 0.974)	
		100	96.85	96.40	85.71	99.28	0.966	<0.0001
			(95.60 to 97.82)	(95.83 to 96.91)	(83.80 to 87.44)	(98.99 to 99.48)	(0.961 to 0.971)	

PPV = positive predictive value. NPV = negative predictive value. AUC = area under the curve.

**Table 7 ijerph-19-07971-t007:** The comparison of smoking statuses and urinary cotinine values according to various ranges of urinary cotinine.

Year	Current Smoking Prevalence (%)	Range of Cotinine (µg/L)	Nonsmoker		Current Smoker		Total	
			n (%)	Concentration (µg/L)	n (%)	Concentration (µg/L)	n (%)	Concentration (µg/L)
2008	22.87%	<12	2732 (62.9%)	2.91 ± 3.39	14 (1.1%)	2.27 ± 4.46	2746 (48.8%)	2.92 ± 3.40
		12–25	739 (17.0%)	17.69 ± 3.67	5 (0.4%)	18.14 ± 4.15	744 (13.2%)	17.71 ± 3.67
		25–50	452 (10.4%)	35.03 ± 7.24	7 (0.5%)	38.89 ± 5.82	459 (8.2%)	35.08 ± 7.22
		50–100	178 (4.1%)	66.95 ± 13.54	17 (1.3%)	80.37 ± 11.84	195 (3.5%)	68.12 ± 13.90
		>100	239 (5.5%)	612.94 ± 615.77	1244 (96.7%)	1260.75 ± 920.26	1483 (26.4%)	1159.58 ± 919.22
		total	4340 (100.0%)	44.99 ± 199.65	1287 (100.0%)	1220.02 ± 930.92	5627 (100.0%)	314.75 ± 691.80
2011	24.95%	<12	1215 (88.4%)	3.19 ± 2.67	18 (3.9%)	2.78 ± 2.82	1233 (67.3%)	3.20 ± 2.70
		12–25	67 (4.9%)	16.74 ± 3.72	4 (0.9%)	16.95 ± 2.77	71 (3.9%)	16.75 ± 3.66
		25–50	24 (1.7%)	34.68 ± 6.50	1 (0.2%)	45.9	25 (1.4%)	35.13 ± 6.74
		50–100	15 (1.1%)	75.19 ± 12.95	8 (1.8%)	76.43 ± 13.99	23 (1.3%)	75.62 ± 13.02
		>100	54 (3.9%)	931.95 ± 725.49	426 (93.2%)	1407.22 ± 930.46	480 (26.2%)	1353.32 ± 921.67
		total	1375 (100.0%)	41.66 ± 229.81	457 (100.0%)	1313.46 ± 963.35	1832 (100.0%)	363.48 ± 761.32
2014	19.08%	<12	3556 (94.7%)	1.61 ± 1.57	11 (1.2%)	3.68 ± 3.92	3567 (76.9%)	1.62 ± 1.60
		12–25	18 (0.5%)	14.86 ± 2.22	4 (0.5%)	14.94 ± 2.04	22 (0.5%)	15.44 ± 2.66
		25–50	13 (0.3%)	34.64 ± 5.87	10 (1.1%)	38.55 ± 10.77	23 (0.5%)	36.59 ± 8.29
		50–100	20 (0.5%)	79.37 ± 14.95	9 (1.0%)	86.34 ± 11.25	29 (0.6%)	82.03 ± 14.11
		>100	147 (3.9%)	889.69 ± 673.21	851 (96.2%)	1352.72 ± 805.83	998 (21.5%)	1288.67 ± 811.61
		total	3754 (100.0%)	36.98 ± 217.51	885 (100.0%)	1302.18 ± 829.72	4639 (100.0%)	286.39 ± 655.64
2018	18.24%	<12	4496 (95.8%)	0.73 ± 0.83	13 (1.2%)	2.95 ± 3.43	4509 (78.6%)	0.74 ± 0.86
		12–25	9 (0.2%)	20.19 ± 4.18	5 (0.5%)	21.08 ± 2.45	14 (0.2%)	20.51 ± 3.57
		25–50	4 (0.1%)	38.32 ± 8.5	4 (0.4%)	40.25 ± 11.15	8 (0.1%)	39.29 ± 9.24
		50–100	14 (0.3%)	74.42 ± 15.86	11 (1.1%)	78.07 ±14.75	25 (0.4%)	75.54 ± 15.08
		>100	169 (3.6%)	881.79 ± 717.80	1014 (96.8%)	1409.46 ± 834.49	1183 (20.6%)	1338.07 ± 841.05
		total	4692 (100.0%)	32.76 ± 213.10	1047 (100.0%)	1366.15 ± 855.65	5739 (100.0%)	277.68 ± 662.80

## Data Availability

The datasets used in this study were obtained from the Korean National Health and Nutrition Examination Survey (KNHANES) between 2008 and 2018. The dataset recruitment process is detailed in the Appendix A.

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
