# Peer review of "Effect of Second-Hand Smoke Exposure on Establishing Urinary Cotinine-Based Optimal Cut-Off Values for Smoking Status Classification in Korean Adults"

_ijerph, 2022, doi:10.3390/ijerph19137971_

Round 1

Reviewer 1 Report

This paper has good stuff in it, but it needs some work to align the findings with the discussion and conclusions.

Two small language issues, based on common use elsewhere in the literature:

1. "Non SHS exposure" should be "No SHS exposure"

2. The two categories of smokers should be re-labeled "Non-daily Smokers" and "Daily Smokers." This is especially important because of the finding that, of the four categories of people in this study (SHS exposure, no SHS exposure, daily smokers and non-daily smokers), it is only the daily smokers whose cotinine levels did not change over the period covered in this study.

There was a two to three order of magnitude difference in cotinine levels, comparing smokers to non-smokers, with no overlap and an opportunity for the authors to pick whatever they chose as a cut-off value they choose between the respective values.

While the no-SHS-exposure group mean cotinine values were about half the level of the SHS-exposure group, as I read the graphic presentations, there is no way to establish a cut-off value to separate these two groups.

While the difference in mean cotinine values between the two groups of smokers were even more, I did not see anything in the graphic presentation that would allow a clean and definitive cut-off value to separate the two.

A major weakness of this study is their failure to fully acknowledge and discuss the huge increase in use of e-cigarettes, snus and other alternative nicotine delivery products between 2008 and 2018. While these products do not present the same health-related risks to users and non-users, when compared to cigarettes, they do generate levels of cotinine similar to cigarettes.

One unanswered question in this study is how people interpreted the questionnaire, given the choice of smoker or non-smoker, without reference to other possible nicotine delivery product.  The data suggest that respondents to the questionnaires interpreted their use of any nicotine delivery product as "smoking," but this is only a suggestion.  Future questionnaires should be expanded to include use of these products with the expectation that many users will use multiple products.

The two major findings of the data presented in this study, as seen by this reviewer, are as follows:

1.  The cotinine levels of daily "smokers" were relatively unchanged through the study period. 

2.  The cotinine levels of the non-daily smokers and non-smokers showed substantial decreases, suggesting substantial reductions in overall use in tobacco and tobacco-related products during the study period.

What these authors found was different from what they expected to find and expected to document in this study. This being the case, the title, abstract, introduction, results, discussion and conclusions narratives need to be re-written to reflect what they found. Since differences in SHS exposure would not allow a clean cut-off value separating those knowingly exposed and not so exposed, the title of the paper should be changed to something like "Trends in Cotinine Levels in Korea, 2008 to 2018"

Reviewer 2 Report

Dear Authors, 

thank you for the possibility to read this artice.

Below you will find some of my suggetions:

- Title - I would propose "adults" not "adult".

- Line 33 - it is well known, that SHS smoking is associated with the risk of COPD - I would advise to add it on the test. 

- Line 41 - not cotinine itself, but cotinine concentration. I woud suggest to change.  The same for lines 44, 47 ........

- Line 50 - I unerstand racial, but what are the cultural factors related to nicotine absorption and metabolism? Explain please and indicate examples.

- Line 59 - in my opinion it should be plural - "are" not "is" - you mention two thigs: flow chart and exclusion criteria. Please, consider.

- Line 356 - doubled "by".

In general, I must admit that although the article is interesting, it's reading  requires a lot of concentration. The multitude of subgroups of respondents causes concerns whether we actually think about a specific group. However, I think that research can be of practical importance and therefore I think that after minor amendments it will be worth publishing. 

Best wishes. 

Reviewer 3 Report

The manuscript titled “Effect of second-hand smoke exposure on establishing urinary cotinine-based optimal cut-off values for smoking status classification in adult” is very interesting for readers and well structured. The aim of this study is important for researches focused on tobacco smoke exposure and its effect on organism. Therefore, this study is novelty, especially the setting of the cut-off value of urinary cotinine to define SHS exposure.

As a reviewer, I have a major point: the qualification to study population was based on the self-reported smoking status. These informations should be verified by determination of cotinine concentration in serum to avoid mistakes in qualification to the study group, e.g. individuals reported non-daily smoking ( whose answered “Yes, I smoke, but not every day.”) in real could be active smokers. Then, in the verified groups should be determined the cotinine in urine.

Round 2

Reviewer 1 Report

While some changes have been made, this paper is not ready for publication. Again, it includes data worthy of publication, but the management and interpretation of these data are deficient in drawing conclusions not fully supported and failing to recognize weaknesses in the data set relative to SHS exposure and use of other nicotine delivery products.

TITLE:  The title references secondhand smoke as a determinant of cotinine values. This is speculative, not firmly established (as discussed below).  The title should be redone to something like “Trends in urinary cotinine values and recommendations for cut-off values for smoking status classification in adults in Korea.”

ABSTRACT:

Lines 14 and 15 reference SHS, but the findings in the paper showed no influence of SHS useable for establishment of cut-off values separating those exposed and not exposed to SHS.

Line 21 states that mean values in non-smokers decreased from 45.0 mcg/L to 32.8 mcg/L during this period. These numbers do not appear in the paper. Table 2 shows the decrease as 5.86 to 0.48.

Lines 26 and 27 states that reductions in SHS exposure caused these reductions. The accurate statement would be that reductions in SHS may have played a role in reduction in cotinine levels in non-smokers. The problem is that the data in Table 2 show a similar level of reduction in persons supposedly not exposed to SHS. Also, the data displayed in Figure 3 show no clear separation of cotinine values, comparing noon-smokers exposed and not exposed to SHS.

CRITICAL ISSUES NOT DISCUSSED IN ABSTRACT:

1.      The reductions in Cotinine values in non-smokers and non-daily smokers is far more than would be expected from the reduction in percentage of persons smoking, suggesting substantial reductions in cigarette consumption among non-daily smokers.

2.      The inability to specify a reliable cut-off value between those exposed and not exposed to SHS suggests that self-reports of exposure to SHS may not be reliable and suggests that the increasing popularity of e-cigarettes and smokeless tobacco during the period under study may have influenced these results.

3.      Future surveys should explore cotinine values in non-smokers using e-cigarettes and smokeless tobacco products.

PRESENTATION OF DATA:

1.       Table 1 should be expanded to separately show the three categories of non-smokers and the two categories of smokers.

2.      The LoD, LoQ, 0.30 and 0.75 data should be presented as a separate table with separate narrative explanation

3.      A new table should be provided, showing the same data for smokers, as shown for non-smokers in Table 2

4.      A separate table should be provided showing what the authors propose as cut-off values, smokers v non-smokers and demonstrating why they chose these values.

5.      The graphics in Figures 2,3 and 4 are excellent. Each should be followed with an explanation of what these figures show and how and why the patterns changed from year to year.

6.      Tables 2, 4 and5, basically showing the data from Figures 2, 3 and 4 can be relegated to the collection of supplemental files

7.      If you have the data, there should be a figure comparing cotinine values in e-cigarette users comparing smokers to non-smokers.

OTHER:

1.      Materials and Methods should show the differences in questions, comparing the different surveys, especially as they relate to e-cigarettes and other nicotine delivery products.

2.      There should be specific discussion and interpretation of the failure of the data to show clear separation between non-smokers exposed and not exposed to SHS.

3.      There should be specific discussion and interpretation of the finding that cotinine values dropped substantially for non-daily smokers, but not for daily smokers.

4.      There should be an expanded discussion of limitations of this study and recommendations for future surveys.

5.      The conclusions and abstract should emphasize the findings noted above.

Reviewer 3 Report

The sentence in lines 350-352 should be changed on: "First, the smoking status verification was not performed by determination of cotinine concentration in serum, which would have allowed to avoid mistakes in qualification to the study group" Please introduce this correction. 
